# The Psychometric Properties and Effectiveness of the Approach-Avoidance Job Crafting Scale among Chinese Kindergarten Teachers

**DOI:** 10.3390/bs13110882

**Published:** 2023-10-25

**Authors:** Xiaoqing Lin, Runkai Jiao, Feifei Li, Di Lu, Hang Yin, Xintong Jiang

**Affiliations:** 1School of Psychology, Northeast Normal University, Changchun 130024, China; linxq799@nenu.edu.cn (X.L.); yinh401@nenu.edu.cn (H.Y.); jiangxt287@nenu.edu.cn (X.J.); 2National Training Center for Kindergarten Principals, Ministry of Education, Changchun 130024, China; 3College of Education, Wenzhou University, Wenzhou 325035, China; 20220053@wzu.edu.cn; 4Medical Humanities Sciences, China Medical University, Shenyang 110122, China; 20232031@cmu.edu.cn

**Keywords:** approach job crafting, avoidance job crafting, kindergarten teachers, psychometric properties

## Abstract

Job crafting is an important concept associated with many positive outcomes, particularly for kindergarten teachers. Specifically, job crafting can play a key role in improving kindergarten teachers’ work experiences and reducing professional dysfunction, considering their stressful work environment. However, the existing scale that integrates the most accepted theories of job crafting—approach–avoidance theory—has not been examined for efficacy in a multicultural context. Thus, in this study, 1273 Chinese kindergarten teachers were selected as subjects to explore the psychological properties of the German version of the Approach-Avoidance Job Crafting Scale (AAJCS) within the Chinese culture context. The sample was randomly divided into two subsamples. In Sample 1 (*N* = 618), item analysis and confirmatory factor analysis (CFA) were conducted. In Sample 2 (*N* = 655), CFA, reliability, and criterion-related validity tests were conducted. The results showed that job crafting has two independent components, each containing four factors, and the third-order model had the best fit. The reliability and validity of the scale were good, suggesting that AAJCS is an appropriate tool for measuring job crafting in a kindergarten population. Future research is needed to test the validity and reliability of the AAJCS on other populations.

## 1. Introduction

Educators significantly contribute to young children’s early development and kindergarten teachers play a crucial role in this process. Given young children’s limited social experience, modest self-care abilities, and rapid physical and cognitive growth, among other unique characteristics, kindergarten teachers’ responsibilities are notably more complex than those of teachers at other educational stages. Simultaneously, kindergarten teachers suffer from low social prestige [1] and low income [2]. These experiences often result in emotional exhaustion, lack of personal fulfillment, and depersonalization at the workplace, ultimately affecting their intent to leave [3]. To improve this situation, scholars have begun to pay more attention to job crafting as a way to improve kindergarten teachers’ work experiences and reduce occupational dysfunction [4,5,6]. Previous research has found that kindergarten teachers who craft their jobs can improve their work engagement [6,7], job satisfaction [4], childcare quality and organizational commitment [5], and work performance [5,6]. Therefore, it is vital to understand the concepts and specific strategies of kindergarten teachers’ job crafting since it can help identify effective ways to improve their work experiences and provide a basis for intervention research.

Job crafting is a form of spontaneous initiative [8,9]. Two main theoretical perspectives underpin the research and analysis of job crafting: the role and resource perspectives. The role perspective originates from the pioneering work of Wrzesniewski and Dutton [9]. This perspective argues that employees take the initiative to change work content and craft job roles to create work identity and meaning [9]. Wrzesniewski and Dutton [9] defined job crafting as an active physical or cognitive change within the boundaries of a task or relation. On the other hand, the resource perspective is rooted in the job demands–resources model. This perspective holds that individuals engage in job crafting to seek a dynamic balance between job demands and resources [8]. Both perspectives emphasize the positive effects of job crafting. However, research has found that some job crafting is detrimental to work engagement and performance [10]. Existing role and resource perspectives fail to account for unfavorable job crafting and contain overlapping components (e.g., both perspectives refer to relational crafting).

To address this challenge, researchers have attempted to integrate these two perspectives [11,12,13]. Given that role and resource perspectives involve direction, the integration of job crafting structures has drawn upon theories such as approach-avoidance theory [14] or regulatory focus theory [15]. Bruning and Campion [11] proposed the concept of job crafting based on approach-avoidance theory and believed that job crafting is an active change in job content. Job crafting can significantly enhance employees’ work efficiency and meaning [11].

Researchers have developed measurement scales based on various theoretical perspectives. However, previous studies have used scales based on either a role or resource perspective to explore kindergarten teachers’ job crafting. Some researchers have used the scale developed by Leana et al. [5] based on the role perspective to measure kindergarten teachers’ job crafting. However, this scale includes the task crafting dimension only. Slemp and Vella-Brodrick [16] developed a scale that was used to evaluate kindergarten teachers’ job crafting. This scale comprises task crafting, relational crafting, and cognitive crafting. In contrast, other researchers used the scale developed by Tims et al. [17] from the resources perspective to measure kindergarten teachers’ job crafting. This scale includes increasing social and structural job resources, increasing challenging job demands, and decreasing hindering job demands. In recent years, scholars have recognized that a single perspective may not comprehensively explain all results. In light of the need for further integration of job crafting research from these two perspectives, scholars have recommended an integrated perspective [11,12]. Zhang and Parker [12] combined all previous job crafting dimensions and proposed a hierarchical model [12]. In the hierarchical model, at the top level, different directions of job crafting are distinguished based on the approach-avoidance theory: approach crafting and avoidance crafting. At the second level, the specific forms of job crafting are distinguished: cognitive crafting and behavior crafting. The resource perspective is located at the bottom of the model, emphasizing different aspects of crafting: resource crafting and demand crafting.

Lopper et al. [18] revised the third-order model proposed by Zhang and Parker [12] and developed the Approach-Avoidance Job Crafting Scale (AAJCS), which, as suggested by the name, includes both approach and avoidance job crafting. It comprises eight dimensions and has proven to be reliable and valid for German adult samples. Approach job crafting emphasizes activities needed to achieve a positive end state, which is positively correlated with work engagement [7,18,19], work meaning [9], and self-efficacy [20]. In contrast, avoidance job crafting emphasizes eliminating the negative end state, which is positively correlated with work engagement [13,18] and work meaning [21] but not with self-efficacy [22].

Job crafting is important for teachers and is of interest to researchers. Scholars have suggested that the scale structure may be different in different cultures, and cross-cultural consistency needs to be further tested [18,23]. However, to the best of our knowledge, the Chinese version of the AAJCS (AAJCS-C) has yet to be translated, revised, and tested in the Chinese context. This study evaluated the reliability and validity of the AAJCS-C based on the research by Lopper et al. [18] within the Chinese context, specifically exploring the scale’s psychometric properties, through which we determined whether it demonstrates consistency in a multicultural environment among kindergarten teachers. Considering the above views, the following hypotheses were proposed in this study: the AAJCS (1) represents a third-order factor model that includes eight dimensions, (2) has sufficient reliability in the Chinese context, and (3) has good validity by selecting work engagement, work meaning, and self-efficacy as criterion variables.

## 2. Materials and Methods

### 2.1. Procedure and Data Collection

Kindergarten teachers were purposefully selected from one to three provinces of each of China’s three economic development regions: the eastern region (e.g., Guangdong Province), the central region (e.g., Hubei Province), and the western region (e.g., Sichuan Province).

The kindergarten principals facilitated the distribution of the online questionnaires to teachers and invited them to participate voluntarily. Before the survey, all the teachers were provided informed consent forms and clear instructions for completing the questionnaire. The questionnaire instructions clarified that participation in the survey was anonymous and that they could withdraw from the survey at any time should they so wish. Ethical approval for this survey was obtained from the Institutional Ethics Committee of the first author’s university.

In total, 1581 questionnaires were collected, 1273 of which were deemed valid. The remaining 308 were excluded because (1) their average response time per item was less than two seconds [24,25] and (2) the participants failed to pass attention checks (e.g., “Please choose the second choice ‘somewhat disagree’ in this item”). Participants with valid responses were then randomly divided into two samples [26]. This approach validated the stability of the structural factor solution across the two subsamples [27].

### 2.2. Participants 

Sample 1 comprised 618 kindergarten teachers with an average age of 32.70 years (SD = 8.29, range 20–55 years) and an average work experience of 10.29 years (SD = 8.38). Concerning education level, 37.7% of the participants had completed high school or postsecondary specialized college education, while 62.3% had completed undergraduate or postgraduate degrees. 

Sample 2 comprised 655 kindergarten teachers with an average age of 31.95 years (SD = 7.70, range 20–58 years) and an average work experience of 8.73 years (SD = 8.07). Regarding education level, 47.50% had completed high school or postsecondary specialized college education, while 52.50% had completed undergraduate or postgraduate degrees.

### 2.3. Measurement

#### 2.3.1. Approach-Avoidance Job Crafting 

The AAJCS contains two subscales: the Approach Job Crafting Scale (AJCS) and the Avoidance Job Crafting Scale (AvJCS). The AAJCS includes forty items measured on a 5-point Likert scale ranging from 1 (totally disagree) to 5 (totally agree). The AJCS and AvJCS each contain four dimensions.

In this study, to preserve the meaning and context of the authors’ version [28], the AAJCS was translated into the AAJCS-C with the consent of the original authors. First, three psychology doctors separately translated the AAJCS into Chinese. The three versions were then discussed and the first draft was developed according to the original meaning and Chinese expression habits. Second, a bilingual doctor translated the Chinese version of the scale into English. Finally, the English translation was compared with the original manuscript, and disputed aspects of this version were discussed, confirming the clarity of all topics and wording.

#### 2.3.2. Work Engagement 

The Work Engagement Scale was developed by Schaufeli et al. [29]. It includes nine items and three dimensions (vigor, dedication, and absorption). An example item is “I am immersed in my work”. The items were rated on a 7-point Likert scale ranging from 1 (strongly disagree) to 7 (strongly agree). In the current study, the Cronbach’s alpha coefficients for vigor, dedication, absorption, and total score were 0.94, 0.95, 0.92, and 0.97, respectively.

#### 2.3.3. Work Meaning

The Work Meaning Scale was developed by Spreitzer [30]. It consists of three items, each rated on a 5-point Likert scale ranging from 1 (strongly disagree) to 5 (strongly agree). An example item is “My job activities are personally meaningful to me”. In the current study, Cronbach’s alpha coefficient was 0.95.

#### 2.3.4. General Self-Efficacy

Wang et al. [31] translated and revised the Chinese version of the General Self-Efficacy Scale. This one-dimensional scale comprises ten items, each of which is rated on a 4-point Likert scale ranging from 1 (completely inconsistent) to 4 (completely consistent). An example item is “I can always solve problems if I try my best”. In the current study, the Cronbach’s alpha coefficient was 0.92. 

### 2.4. Statistical Analysis

The Statistical Package for the Social Sciences (SPSS) version 21.0 was used for item analysis, criterion validity, and reliability analysis. Mplus version 8.3 was used for the confirmatory factor analysis (CFA).

Sample 1 involved a two-step data analysis. Initially, item analysis was performed to assess item validity using item-total correlations. We then determined whether each item on the scale had sufficient discriminative power. An independent-sample *t*-test was employed to compare the items’ 27% upper and lower groups, and a significant difference (*p* < 0.05) indicated good item discrimination [32]. In the next step, the factor structure of the AAJCS-C was analyzed using the robust maximum likelihood method (MLR). The fit measures included the χ^2^ statistic, the chi-square ratio (χ^2^/df), the comparative fit index (CFI), the Tucker–Lewis index (TLI), the root-mean-square error of approximation (RMSEA), and the standardized root-mean-square residual (SRMR). The alternative fit indices were used because χ^2^ values are significant in large samples [33]. Thus, the fit was assessed based on four criteria: CFI ≥ 0.90, TLI ≥ 0.90, RMSEA ≤ 0.08, and SRMR ≤ 0.08 [34]. Moreover, all models were compared using the Akaike Information Criterion (AIC) and Bayesian Information Criterion (BIC) [35]. These criteria aid in evaluating the strengths and weaknesses of the competitive models, with smaller values indicating a better fit [36]. 

Sample 2 was subjected to a three-step data analysis. First, the factor structure of the AAJCS-C was re-examined using MLR to validate the scale model. Second, reliability tests were conducted to assess the internal consistency of AAJCS-C. Third, the relationships between the AAJCS-C and other variables (i.e., working engagement, work meaning, and self-efficacy) were analyzed to examine concurrent validity.

## 3. Results

### 3.1. Item Analysis

First, the total score for the Chinese versions of the Approach Job Crafting Scale (AJCS-C) and Avoidance Job Crafting Scale (AvJCS-C) were evaluated separately. The correlations between each item and the scales’ total score were examined to determine item validity. Correlation analysis showed that the correlation coefficient between the AJCS-C total score and its items was 0.75–0.89, *ps* < 0.001. The correlation coefficient between the AvJCS-C total score and its items was 0.45–0.82, *ps* < 0.001. Second, the total scale scores were sorted into upper and lower groups. A *t*-test was conducted for each item in both groups. All items showed significant differences between the upper and lower groups (*p* < 0.001). The correlation results are summarized in Table 1.

### 3.2. Confirmatory Factor Analysis (CFA)

As the AAJCS was developed based on a theoretical hypothesis structure, it was more appropriate to use CFA to directly examine the model and factor loadings for each item. In Sample 1, CFA was used to assess the fit indices of the eight-dimensional measurement model and five competing models: the one-factor, bifactor (resources or demands), bifactor (cognitive or behavioral), second-order, and third-order models. The results are summarized in Table 2. Each measurement model exhibited an acceptable or moderate fit, confirming the factorial validity of the eight dimensions. Regarding the factor structure of the entire scale, the results showed that the fit indices of the one-factor and bifactor models were unsatisfactory. However, the second- and third-order models achieved recommended fit values. Given that the third-order model had the smallest AIC and BIC values, it was considered to be the most appropriate choice [35]. The CFA also served as the basis for item selection. The factor loadings for the AJCS-C items ranged from 0.80 to 0.95, while for the AvJCS-C, all item factor loadings ranged from 0.60 to 0.90 except for t30 (factor loading = 0.37, “I am able to complete tasks that require me to make tough decisions faster”). The factor loadings for all items are shown in Table 1.

Sample 2 was also analyzed using CFA to further investigate the stability of the third-order factor model. The results indicated that item factor loadings ranged from 0.32 to 0.90. Furthermore, the results of competing models showed that fit indices of the third-order model were favorable and reached recommended standards (χ^2^/df = 2131.73/727, *p* < 0.001, RMSEA = 0.05, SRMR = 0.06, CFI = 0.91, TLI = 0.91). In sum, the third-order AAJCS-C model was the most appropriate. Table 2 presents the fit indices.

### 3.3. Reliability

Sample 2 was used to assess the reliability of the scales. The Cronbach’s alpha coefficients of the AJCS-C and its subdimensions (subdimensions 1 to 4) were 0.96, 0.92, 0.92, 0.94, and 0.92, respectively. The Cronbach’s alpha coefficients of the AvJCS-C and its subdimensions (subdimensions 5–8) were 0.94, 0.88, 0.81, 0.92, and 0.91, respectively. These results indicate that the AAJCS-C had good reliability.

### 3.4. Criterion-Related Validity

Correlations between approach and avoidance job crafting and criterion-related variables were calculated to examine the concurrent validity of the AAJCS-C. The results showed that approach job crafting was positively correlated with work engagement, work meaning, and self-efficacy. In contrast, avoidance job crafting was negatively correlated with work engagement and work meaning but not with self-efficacy. The results are shown in Table 3.

## 4. Discussion

Kindergarten teachers face internal pressure to promote children’s healthy growth and stimulate their professional development. Simultaneously, they grapple with external environmental pressures including those related to social expectations, role conflicts, and income. These pressures cause them to experience high workloads as well as decreased productivity, job dissatisfaction, motivation, and enthusiasm [3]. Therefore, it is important to understand how teachers craft their work to support them in their educational contexts and design appropriate interventions. This study aimed to analyze the psychometric properties of the AAJCS among Chinese kindergarten teachers. The scale was revised by Lopper et al. based on the hierarchical model of job remodeling proposed by Zhang and Parker [12]. The AAJCS includes both approach and avoidance job crafting. First, the item analysis results indicated that these items had good discriminative power. Therefore, we agree with Lopper et al. [18] that it is appropriate to retain all forty items.

To gain further insight into the relative contributions and significance of the factors, we conducted CFA, and the results revealed that the factor loadings of the approach job crafting items were above 0.80, indicating strong associations. The factor loadings were generally above 0.40 for avoidance job crafting items, except for the avoidance behavioral demands crafting item “I get tasks that require me to make tough decisions done fast.” This item had a factor loading of 0.37, which was slightly below the 0.40 threshold. Lopper et al. [18] believe that this item aligns with the criteria for avoidance behavior, supporting the idea that making difficult choices necessitates a certain level of effort, while completing tasks more efficiently demonstrates a deliberate avoidance attitude. A similar perspective is presented by Tims and Bakker [17]. Specifically, the decreasing hindering job demands dimension suggests that individuals attempt to avoid making difficult decisions at work. Employees may achieve this by promptly completing tasks, avoiding extended contemplation, or making difficult decisions. Consequently, we respect the original authors’ decision to retain this item.

In Sample 1, we used five CFA models to evaluate the factor structure of the AAJCS-C. The findings indicated that the second- and third-order models fit better than the one-factor and bifactor models. Notably, the CFI, RMSEA, and SRMR values of the second- and third-order models met the recommended model-fitting criteria. Consistent with Lopper et al.’s research, we used the Akaike information criterion (AIC) and Bayesian information criterion (BIC) as alternative indicators [35]. The results corroborated the superiority of the third-order model, which aligns with Lopper et al.’s findings. In addition, based on the three-level hierarchical model, in which the top level encompasses approach or avoidance job crafting, our data supported the notion that the second-order model could be further subdivided into the third-order model. This alignment between the data and theoretical model was further substantiated by the consistency of the results obtained for Sample 2. The results from Sample 2 also demonstrated superior fit indices for the third-order model, affirming the appropriateness of this model for representing the factor structure of the AAJCS-C. Thus, we believe that the third-order model better fits the factor structure of the AAJCS-C. 

Furthermore, the reliability analysis revealed the strong internal consistency of the AAJCS-C, providing further evidence of its robustness. The criterion-related validity analysis revealed that the AJCS-C was positively associated with work meaning and work engagement, whereas the AvJCS-C exhibited negative correlations with work meaning and work engagement, consistent with the results of previous studies [11,18]. Compared with avoidance job crafting, approach job crafting emphasizes taking the initiative to deal with external challenges, in which individuals actively adjust tasks and relationships to better suit their abilities and preferences. This sense of fit increases individual job engagement and enhances the meaning of work. Additionally, our study identified a positive correlation between approach job crafting and self-efficacy, in line with previous research results [20]. However, avoidance job crafting did not have a significant relationship with self-efficacy. Scholars found that decreasing demands (i.e., avoidance job crafting) were not related to self-efficacy, while interventions in approach job crafting could improve self-efficacy [22]. These findings were consistent with those obtained using the German version of the AAJCS, supporting the conclusion that the AAJCS-C has good reliability and validity among Chinese kindergarten teachers.

This study provides an effective and reliable tool for evaluating kindergarten teachers’ job crafting, which will be helpful for the implementation of intervention programs. Scholars can implement an intervention plan based on the measurement results of the AAJCS and understand the changes before and after the intervention to make the intervention plan more realistic. In addition, it can help kindergarten teachers understand the degree of adjustment and improvement in their work and then adjust their job crafting methods in a targeted way to increase self-efficacy, work engagement, and work meaning. Employers have a duty to protect their employees [37]. Therefore, understanding kindergarten teachers’ attitudes and levels of job crafting can help managers take action to increase proactive behaviors, which in turn can improve the work experience.

This study has two limitations stemming from the specific characteristics of job crafting among kindergarten teachers. First, our sample exclusively comprised kindergarten teachers, which may have restricted the generalizability of our findings to other groups. Future studies should increase the diversity of the sample and verify the AAJCS-C in other groups. Second, the data were collected based on self-reported answers, which may not accurately reflect objective behavior. Therefore, future studies should adopt a mixed-methods approach that combines qualitative and quantitative methods. By incorporating interviews and questionnaires, researchers can triangulate the results and gain a more comprehensive understanding of the structure and nature of job crafting among kindergarten teachers. Despite these limitations, this study provides valuable insights into the psychometric properties of the AAJCS-C and its applicability to Chinese kindergarten teachers. Further research on these limitations could contribute to a more comprehensive understanding of job crafting among different groups.

## 5. Conclusions

The present study revised and examined the validity the AAJCS in the Chinese culture. The results confirm that the AAJCS is a reliable and effective tool for measuring kindergarten teachers’ job crafting in the context of the Chinese culture. Further, they suggest that kindergarten teachers’ different types and levels of job crafting can be clearly understood through this scale. Future research could verify the effectiveness of the scale in different groups or cultures.

## Figures and Tables

**Table 1 behavsci-13-00882-t001:** The factor loadings and correlation coefficient of the Approach-Avoidance Job Crafting Scale (Sample 1, *N* = 618).

Dimension	Item	r	Factor Loading
Approach Behavioral Resources Crafting	I proactively establish relationships with other people (e.g., colleagues, principal, parents of children) at work.	0.90 ***	0.86
	I will proactively seek work feedback from others.	0.89 ***	0.84
	I proactively work with people (e.g, colleagues) who get along well.	0.91 ***	0.88
	I actively explore tasks that can use my skills.	0.91 ***	0.89
	I actively explore tasks that can help improve my professional level.	0.84 ***	0.82
Approach Behavioral Demands Crafting	I will take on new tasks proactively when I don’t have much work to do.	0.90 ***	0.85
	I will proactively take on additional tasks at work.	0.91 ***	0.86
	I invest extra time and energy to better complete tasks.	0.87 ***	0.85
	I actively seek challenges at work.	0.90 ***	0.88
	I am willing to take on more responsibilities at work.	0.91 ***	0.89
Approach Cognitive Resources Crafting	I can realize how my work help me improve my professional skills.	0.91 ***	0.90
	I think that my work is very important for social development.	0.90 ***	0.85
	I think that I can improve myself by collaborating with people (e.g., colleagues, principal, parents of children).	0.93 ***	0.90
	I think my work has an important meaning in my life.	0.93 ***	0.91
	I focus on the positive aspects of my work.	0.93 ***	0.90
Approach Cognitive Demands Crafting	I think difficult tasks as a positive challenge.	0.91 ***	0.88
	I think that difficult tasks can help me improve my professional abilities.	0.94 ***	0.93
	I consider tasks with significant responsibility to be a challenge for me.	0.95 ***	0.94
	I think that difficult decisions at work can help me improve my professional abilities.	0.95 ***	0.95
	I think I can complete tasks better under time pressure.	0.88 ***	0.80
Avoidance Behavioral Resources Crafting	I spend less time on tasks that don’t interest me.	0.85 ***	0.79
	My enthusiasm will decrease if the tasks do not help my professional development.	0.87 ***	0.83
	I will delegate tasks with less responsibility to others.	0.79 ***	0.69
	I will give a lower priority to tasks if I do not receive feedback.	0.89 ***	0.87
	I invest less time in tasks that don’t require me to use my skills.	0.85 ***	0.83
Avoidance Behavioral Demands Crafting	I will put tasks that are too demanding at the end of the line.	0.74 ***	0.63
	I will delegate tasks I don’t like to others.	0.80 ***	0.78
	I will postpone completing unnecessary tasks.	0.83 ***	0.81
	I will delegate tasks that may conflict with others (e.g., colleagues, principal, parents of children).	0.75 ***	0.71
	I will respond more quickly to tasks involving difficult decisions.	0.55 ***	0.37
Avoidance Cognitive Resources Crafting	I try to avoid thinking about tasks that don’t require me to make decisions.	0.84 ***	0.77
	I try to avoid thinking about tasks that don’t help me in my professional development.	0.88 ***	0.83
	I try to avoid spending too much thought on tasks for which I don’t get any support.	0.90 ***	0.87
	I try to avoid taking seriously tasks that don’t use my skills.	0.89 ***	0.89
	I try to avoid thinking about tasks that do not allow me to make progress at work.	0.90 ***	0.89
Avoidance Cognitive Demands Crafting	I try to avoid thinking about tasks that make me emotionally nervous.	0.81 ***	0.81
	I rarely think about tasks that I don’t like.	0.88 ***	0.85
	I try to avoid thinking about tasks that require me to make difficult decisions.	0.90 ***	0.90
	I rarely imagine that I need to work with difficult people (e.g., colleagues, principal, parents of children).	0.76 ***	0.62
	I rarely think about stressful tasks.	0.84 ***	0.75

Note: Standardized factor loadings refer to the proposed job crafting dimension from the CFAs. *** *p* < 0.001.

**Table 2 behavsci-13-00882-t002:** Fit indices for the confirmatory factor analysis validation mode.

Models	χ^2^/df	CFI	TLI	RMSEA (CI 90%)	SRMR	AIC	BIC
Sample 1 (*N* = 618)							
Measurement models							
ABR	12.36/3 ***	0.99	0.97	0.07	0.01	4802.49	4877.74
ABD	13.92/4 ***	0.99	0.97	0.06	0.01	5238.19	5309.01
ACR	7.38/5 ***	0.99	0.98	0.03	0.01	4021.34	4087.74
ACD	18.06/5 ***	0.99	0.97	0.07	0.01	4293.64	4360.04
AvBR	10.74/4 ***	0.99	0.98	0.05	0.01	7894.10	7964.92
AvBD	10.00/5 ***	0.99	0.98	0.04	0.02	8652.28	8718.68
AvCR	14.79/4 ***	0.98	0.97	0.07	0.02	6919.54	6990.36
AvCD	29.69/8 ***	0.97	0.95	0.07	0.03	9029.56	9113.67
Structural model							
One-factor model	9346.34/740 ***	0.50	0.47	0.14	0.23	56,342.54	56,873.72
Bifactor model (resources or demands)	7072.10/734 ***	0.64	0.61	0.12	0.23	53,075.86	53,633.60
Bifactor model (cognitive or behavioral)	6959.41/734 ***	0.64	0.62	0.12	0.22	52,966.11	53,523.85
Second-order model	2015.73/726 ***	0.93	0.92	0.05	0.06	46,160.60	46,753.75
Third-order model	2016.98/727 ***	0.93	0.92	0.05	0.06	46,158.65	46,747.37
Sample 2 (*N* = 655)							
Measurement models							
ABR	13.10/4 ***	0.98	0.97	0.06	0.02	5054.53	5126.29
ABD	11.14/4 ***	0.99	0.98	0.05	0.02	6007.50	6079.26
ACR	15.59/4 ***	0.99	0.97	0.07	0.01	4788.92	4860.68
ACD	14.41/3 ***	0.98	0.95	0.08	0.01	5775.03	5851.27
AvBR	11.49/4 ***	0.99	0.98	0.05	0.02	8417.21	8488.96
AvBD	18.93/4 ***	0.98	0.94	0.08	0.03	8940.93	9012.69
AvCR	5.80/4 ***	0.99	0.99	0.03	0.02	7553.82	7625.58
AvCD	25.92/8 ***	0.98	0.97	0.06	0.02	9326.83	9412.04
Structural model							
Second-order model	2131.41/726 ***	0.91	0.91	0.05	0.06	51,617.61	52,218.55
Third-order model	2131.73/727 ***	0.91	0.91	0.05	0.06	51,615.83	52,212.28

Note: ABR, approach behavioral resources crafting; ABD, approach behavioral demands crafting; ACR, approach cognitive resources crafting; ACD, approach cognitive demands crafting; AvBR, avoidance behavioral resources crafting; AvBD, avoidance behavioral demands crafting; AvCR, avoidance cognitive resources crafting; AvCD, avoidance cognitive demands crafting. *** *p* < 0.001.

**Table 3 behavsci-13-00882-t003:** The criterion validity of AAJCS-C.

Construct	Mean	SD	1	2	3	4	5
1. Approach crafting	4.36	0.64	1				
2. Avoidance crafting	2.62	0.81	−0.20 ***	1			
3. Work engagement	5.78	1.14	0.61 ***	−0.33 ***	1		
4. Work meaning	4.27	0.78	0.57 ***	−0.30 ***	0.86 ***	1	
5. Self-efficacy	3.01	0.50	0.42 ***	−0.04	0.45 ***	0.43 ***	1

Note: *** *p* < 0.001.

## Data Availability

The data that support the findings of this study are contained within the article and available from the corresponding author upon reasonable request.

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
