# Peer review of "The Psychometric Properties and Effectiveness of the Approach-Avoidance Job Crafting Scale among Chinese Kindergarten Teachers"

_behavsci, 2023, doi:10.3390/bs13110882_

Round 1

Reviewer 1 Report

Comments and Suggestions for Authors

The peer-reviewed scientific study is undoubtedly a very interesting scientific work, the content of which can be beneficial not only for theory but also for practice. At the outset, I can state that it has considerable potential.

The authors probably neglected to read enough the instructions for authors available on the journal's website and strictly follow them so that the content structure is respected.

The introduction lacks a clearly stated reason for writing this scientific study, a really clearly stated main goal and secondary goals.

Part of the introduction is also an incorrectly mandatory part "Theoretical overview", which should be a separate part. I also consider the composition of the used literature as a problem. I understand that domestic literature must objectively prevail, but in order to expand knowledge, I recommend supplementing the Theoretical overview and thus also the list of used literature with works by other than only domestic authors. Authors such as:

Tokareva, V; Davydova, I and Adamova, E. 2021. Legal problems of the use of orphan works in the digital age, JURIDICAL TRIBUNE-TRIBUNA JURIDICA 11 (3), pp.452-471, doi:

Marius EZER. 2020. THE JOINT INVESTIGATION PROCEDURE OF WORK ACCIDENTS. Perspectives of Law and Public Administration, (9), SI, pp. 91-94

Peracek, T. 2020. HUMAN RESOURCES AND THEIR REMUNERATION: MANAGERIAL AND LEGAL BACKGROUND, paper presented at 13th International Scientific Conference on Reproduction of Human Capital - Mutual Links and Connection (RELIK), pp.454-465, Date NOV 05-06, 2020, Prague, Czech Republic.

I did not find established hypotheses, only clearly defined research questions, which the authors should have clearly answered in the final chapter, which would fulfill the very meaning of their scientific work. These unequivocal answers are missing. The conclusion is so short that it doesn't even have to be in that form. It clearly needs to be expanded.

As I mentioned, in the conclusion it is necessary to clearly answer the established research questions and hypotheses with proper justification. It is not the purpose of the conclusion to describe what the article was about.

Reviewer 2 Report

Comments and Suggestions for Authors

This study evaluates the external validity of AAJCS in Chinese kindergarten teachers, showing that the three-order model presents the best fit compared to the other models and is robust against cultural variation and diversities.

Comments on the Quality of English Language

Please refer to the enclosed document for minor comments.

Reviewer 3 Report

Comments and Suggestions for Authors

I am glad to have this opportunity to review this article. After reading it, I appreciate great efforts made by the author(s). However, there is one problem. Hence, this article must have a minor revision. The problem is listed as follow for your reference.

1.     This article is interesting.

2   This paper must emphasize why kindergarten teachers should be the research object.

Round 2

Reviewer 1 Report

Comments and Suggestions for Authors

I agree with publication this version.

Author Response

Dear reviewer,

Thank you very much for your recognition of our manuscript.

Warm regards,

23 Oct 2023

Reviewer 2 Report

Comments and Suggestions for Authors

See enclosed

Comments on the Quality of English Language

See enclosed
